# Treatment of Hodgkin Lymphoma Relapsed after Autologous Stem Cell Transplantation

**DOI:** 10.3390/jcm9051384

**Published:** 2020-05-08

**Authors:** Eva Domingo-Domènech, Anna Sureda

**Affiliations:** Department of Hematology, Catalan Institute of Oncology, Hospital Duran i Reynals, IDIBELL, University of Barcelona (UB), L’Hospitalet de Llobregat, 08907 Barcelona, Spain; asureda@iconcologia.net

**Keywords:** relapsed/refractory Hodgkin’s Lymphoma, autologous stem cell transplantation, brentuximab vedotin, checkpoint inhibitors, allogeneic stem cell transplantation

## Abstract

Although autologous stem cell transplantation (auto-HCT) is the standard of care for patients with refractory/relapsed (R/R) classical Hodgkin’s lymphoma (cHL), there is still a significant proportion of patients that relapse after the procedure. This review contemplates different treatment strategies for patients with cHL that relapse or progress after auto-HCT. Allogeneic stem cell transplantation (allo-HCT) has, for many years, been the only curative option for this group of patients. Although the advent of haploidentical donors has allowed for the possibility to allograft almost all patients that are in need of it and to eventually improve historical results, allo-HCT is still associated with substantial morbidity and mortality. Brentuximab vedotin (BV) is an antibody drug conjugate that binds to CD30 antigen; BV is able to give up to 34% metabolic complete remissions (mCR) in HL patients that fail auto-HCT. Unleashing the immune system with PD-1 inhibitors has resulted in remarkable responses in a number of malignancies, including HL. Nivolumab and pembrolizumab offer a 20%–25% mCR and 40%–50% partial remissions, with an acceptable safety profile. R/R cHL do have several options nowadays that, without any doubt, have significantly improved the long-term outcome of this hard-to-treat population.

## 1. Introduction

Most patients with classical Hodgkin’s lymphoma (cHL) can be cured with conventional chemotherapy (CT) ± radiotherapy (RT) [1,2]. Nevertheless, the prognosis of those patients who present with primary refractory disease, or those patients who despite achieving a first complete remission (CR), eventually relapse, remains poor. Autologous hematopoietic stem cell transplantation (auto-HCT) is the standard of care for the majority of relapsed/refractory (R/R) cHL patients on the basis of two randomized trials that showed significant benefits for auto-HCT over conventional CT for relapsed diseases [3,4]. Although a significant proportion of patients can be cured with high-dose therapy and auto-HCT, up to 50% of them still relapse. There is still little information on the predictors of outcomes for patients whose disease recurs after auto-HCT. A retrospective analysis of the Lymphoma Working Party (LWP) of the European Group for Blood and Marrow Transplantation (EBMT) and Gruppo Italiano Trapianto di Midollo Osseo (GITMO) [5] indicates that overall survival (OS) of patients that relapse after auto-HCT is of only 32% at 5 years; independent risk factors for OS were having a stage IV and bulky disease at the time of disease recurrence, early relapse (<6 months), inadequate performance status, and being aged ≥50 years. Relapse after auto-HCT still represents a clear unmet medical need. Allogeneic stem cell transplantation (allo-HCT) has, for many years, been the only curative option for these patients [6]. Nevertheless, the landscape of R/R cHL treatment has changed significantly since the approval of brentuximab vedotin (BV) in 2011 for the treatment of patients who are not candidates for auto-HCT after at least two lines of chemotherapy and for treating patients that fail auto-HCT [7]. More recently, the approval of checkpoint inhibitors (CPIs), pembrolizumab, and nivolumab [8,9] has further increased the treatment options in this setting. Other novel agents under investigation include histone deacetylase (HDAC) inhibitors, Janus kinase 2 (JAK2) inhibitors, and immunomodulators, as well as early approaches with chimeric antigen receptor (CAR) T cells. In this review, we intend to summarize the recent advances in therapy for patients with cHL who fail auto-HCT.

## 2. Antibody-Based Therapies

### 2.1. Brentuximab Vedotin

Brentuximab Vedotin (BV) is a conjugated anti-CD30 monoclonal antibody that was approved in 2011 by both the FDA and European Medical Agency (EMA) for the treatment of patients that relapse after auto-HCT or that did not respond to two prior lines of chemotherapy and were not considered suitable for an auto-HCT. The phase II prospective pivotal trial evaluated the efficacy and safety of BV (1.8 mg/kg IV every 3 weeks up to 16 cycles) in 102 patients with relapsed or refractory cHL after auto-HCT [7]. The overall response rate (ORR) was 75%, and 34% of the patients achieved a metabolic CR (mCR). The median progression-free survival (PFS) was 5.6 months, and the median duration of response for those patients that achieved a CR was 20.5 months. The OS rate was 41%, and PFS was 22% after 5 years of follow-up [10]. Those patients that achieved CR with BV had estimated OS and PFS rates of 64% and 52%, respectively. Despite efficacy with BV monotherapy, most patients required additional therapy within one year. Unfortunately, the low number of patients included in the pivotal trial precluded the identification of prognostic factors that could identify a population of patients that were able to achieve a durable CR and long-term disease control without the addition of other treatment strategies.

BV is, overall, a very well-tolerated drug, and the toxicity profile has been quite constant throughout the different trials where the drug has been tested. The most frequently seen treatment-related adverse events (AEs) were peripheral sensory neuropathy (42%), nausea (35%), vomiting (13%) and diarrhea (18%), fatigue (34%), neutropenia (19%), pyrexia (14%), arthralgia (12%) and myalgia (11%), pruritus (12%), peripheral motor neuropathy (11%), and alopecia (10%). Fifty-six patients (55%) experienced adverse events ≥ grade 3, and 12 patients had adverse events that led to treatment discontinuation, the most common being sensory neuropathy.

BV has an increasing role in first-line treatment [11], in combination with salvage chemotherapy before auto-HCT [12,13] and as consolidation therapy after autologous transplantation [14,15], so its use in the relapse setting will be modified in the next years. Most of the patients will be exposed to BV before auto-HCT, and the possibility for them to receive a second course of BV can be a viable option. Retreatment with BV monotherapy was investigated in patients with cHL or systemic anaplastic large cell lymphoma (ALCL) that relapsed after achieving CR or a partial remission (PR) with initial BV therapy [16]. The ORR was 60% (30% mCR) in HL patients. The median duration of response for those patients that achieved an objective response was 9.5 months. Of the 19 patients with an objective response, seven patients had not progressed or died at the time of study closure; the duration of response ranged from 3.5 to 28 months. Of the patients that achieved a CR (n = 11), 45% had a response duration over one year. AEs were quite similar in type and frequency in re-treated patients in comparison to those observed in the pivotal trials of BV monotherapy, with the exception of peripheral neuropathy which, as expected, was higher, being grade ≥3 in 48% of the cases.

### 2.2. AFM13

AFM13 is a CD16A/CD30 tetravalent, bispecific antibody that stimulates innate immune cells, such as natural killer (NK) cells and macrophages. It binds CD16A on innate cells to CD30 on HL cells, acting as a bridge to recruit and activate innate immune cells in close proximity to tumor cells [17]. Preclinical data demonstrate very efficient antitumoral activity through the engagement of NK cells. The Phase 1 dose-escalation study [18] recruited 28 patients with relapsed/refractory cHL that received AFM13 at doses ranging from 0.01 to 7 mg/kg. Adverse events were mild to moderate, and the maximum tolerated dose was not reached. Three out of 26 evaluable patients achieved partial remission (11.5%), and 13, stable disease (50%), with a disease control rate (DCR) of 61.5%. This compound was also effective in BV–refractory patients. The ORR was 23% and the disease control rate was 77% in those 13 patients that received doses of ≥1.5 mg/kg AFM13. AFM13 treatment was associated to significant NK-cell activation and a decrease of soluble CD30 in peripheral blood. AFM13 has also been combined with pembrolizumab in a Phase Ib study that included 30 BV refractory HL patients [19]. The most common related AEs were infusion-related reactions (80%), rash (30%), pyrexia (23%), nausea (23%), and diarrhea (20%). To highlight, grade 3 Guillains’ Barre syndrome/radiculopathy was observed in 7% of the patients. The ORR and CR rate for evaluable patients treated at the dose and schedule chosen for the expansion cohort were 87% and 35% by the investigator-based evaluation, respectively. Independent assessment resulted in an ORR of 87% and CR rate of 39% for these patients.

### 2.3. ADCT-301

ADCT-301 is an antibody-drug-conjugate composed of human IgG1 HuMax-TAC against CD25, that is conjugated to a pyrrolobenzodiazepine dimer warhead with a drug–antibody ratio of 2:3. In a Phase 1, open-label, dose-escalation and dose-expansion study that included patients with relapsed/refractory cHL, ADCT-301 was able to achieve an ORR of 81% and a CR rate of 50% [20]. The median duration of response and PFS were 7.7 and 6.7 months, respectively. The most common treatment-associated AEs were liver function abnormalities, cutaneous rash (13.3%), anemia (8.3%), and thorombocytopenia (5.0%). There is a currently ongoing Phase II clinical trial to confirm these preliminary positive data.

## 3. Checkpoint Inhibitors

Reed Sternberg cells (HRS) are rare primary malignant cells, surrounded by a background of mixed inflammatory and immune cells that infiltrate the lymph node. In addition to CD30 expression on the HRS cells, another characteristic of HL pathogenesis is the maintenance of the adequate immune microenvironment that allows for the proliferation and survival of HRS cells. Chromosomal analyses of HRS cells have demonstrated the frequent presence of 9p24.1 amplification, which leads to upregulation of PD-1 ligands and JAK2 [21,22]; this makes PD-1 inhibitors an ideal therapy for HL.

Nivolumab and pembrolizumab are the two anti-PD-1 antibodies that have been approved for treatment of multiple relapsed/refractory HL. Nivolumab single-agent treatment for relapsed/refractory HL was tested in the large Phase II trial CHECKMATE 205 that included 243 patients [8]. Nivolumab was administered at a dose of 3 mg/kg every 2 weeks until disease progression or toxicity, to three cohorts of patients: cohort A (*n* = 63), BV naïve patients; cohort B (*n* = 80), patients treated with BV after auto-HCT; and cohort C (*n* = 100), patients who received BV before and/or after auto-HCT. The ORR for all patients was 69%, with 16% patients achieving a mCR. Median PFS was 14.7 months, being almost double in those patients that achieved a CR (22.2 months). Notably, even patients with SD had a median PFS of 11 months and 1-year OS of 96%, similarly to patients who achieved PR or CR. The 1-year OS rate was 92% for all patients, and 59% for those patients with progressive disease. The survival curves were similar for all the cohorts, as well as for all deepnesses of response, demonstrating its benefit even in patients with stable disease. Of the 130 patients that progressed, 80 were treated beyond progression (median of 11 more doses of nivolumab), with clinical benefits in 55% of them (median time to next treatment of 20 months and a 2-year OS of 87%) [21]. Similar data have been reported from the Phase II study of pembrolizumab, KEYNOTE-087 [9]. Pembrolizumab was administered at a flat dose of 200 mg every 3 weeks in 210 patients. This study also included three cohorts: cohort 1 (*n* = 69), with patients who relapsed after auto-HCT and BV; cohort 2 (*n* = 81), patients that received previous BV but were ineligible for auto-HCT; and cohort 3 (*n* = 60), BV naïve progressing after auto-HCT patients. The ORR was 71.9%, with 27.6% mCR, at a median follow-up of 27.6 months. The median duration of response was 16.5 months in all patients, higher in cohorts 1 and 3, while the median PFS not reached in CR patients was of 13.8 months in PR patients and of 10.9 months for those with stable disease. Of the 151 responders, 42.5% had a response of more than 24 months, and 24.5% had ongoing responses. Median overall survival was not reached in all patients, nor in any cohort. Immune-related AEs were of special interest with checkpoint inhibitors, the most frequent organ affected being the thyroid gland (12–13.8% of the patients), with AEs leading to treatment discontinuation in 6.7% of patients [8,9,23,24,25].

Combination immunotherapies have shown promising results. The combination of nivolumab plus BV has been evaluated as first salvage therapy in relapsed/refractory cHL patients, followed by auto-HCT in a Phase I/II clinical trials. Patients received four cycles of the combination with an ORR of 82% and a 61% CR, it being the most common AE infusion-related reactions in 44% of the patients [26]. Other combinations of checkpoint inhibitors with platinum-based schemes in first relapse are actually under investigation. The Phase I E4412 trial combined a tumor cell-targeting drug, such as BV, with a CPI with the objective of activating the immune cells of the tumor’s microenvironment to treat patients with relapsed/refractory cHL. The ORR in those patients treated with BV and ipilimumab, a monoclonal antibody against cytotoxic T lymphocyte antigen 4 (CTLA-4), was of 67%, and the CR rate was of 55% in this heavily pretreated population of patients [27]. The combination of BV and nivolumab gave an ORR of 95% and a CR rate of 65% [28]. The last group of patients included in the trial received the triple combination of BV, nivolumab, and ipilimumab; the ORR was of 95% and the CR rate of 84%, but at the expense of an increased grade 3 immune (including one diabetes, one pancreatitis, and one Steven Johnson syndrome) or worse AEs, compared with the BV plus nivolumab doublet [29]. Nevertheless, and in general, the treatment was generally well-tolerated in all arms, but there were two cases of grade 5 pneumonitis in nivolumab-containing arms. Neither median PFS nor OS were reached with a median follow-up of 0.52 years and 0.82 years, respectively. A follow-up randomized phase II trial comparing the two most effective arms, BV and nivolumab vs. BV, nivolumab, and ipilimumab, is ongoing at this point in time.

### 3.1. Bendamustine

The bifunctional alkylator bendamustine hydrochloride is considered an attractive agent to be used in this clinical setting because of a variety of mechanistic differences compared to other alkylating agents employed in cHL. The largest clinical study [30] included 67 HL patients that had failed either an auto-HCT (*n* = 45, 67%) or an autologous/allogeneic (*n* = 22, 33%) HCT. Bendamustine was administered as a single drug at 90 mg/m^2^ iv or 120 mg/m^2^ iv on days 1 and 2 of each 28-day cycle for a median of three cycles. The ORR was 57%, with a 1-year PFS of 49% and a 1-year OS of 70%. Similar data was observed in a Phase II clinical trial with monotherapy bendamustine at a dose of 120 mg/m^2^ for 2 days every 28 days, which included heavily treated HL patients (median of four lines of treatment). The ORR was 53% (33% CR), with a median duration of response of 5 months, and 20% of them bridging to allo-HCT [31]. Additional studies looking at the combination of bendamustine and gemcitabine have showed a 69% ORR, with a 46% CR in heavily pretreated cHL patients [32]. Nevertheless, the most promising data has been the combination of bendamustine plus BV that is associated to an ORR of 78% in a Phase I/II study that included heavily pre-treated cHL patients [33], and also serves as a very effective and well-tolerated bridging therapy for those patients in need of auto-HCT [34].

### 3.2. Histone Deacetylase Inhibition

Panobinostat, a potent pan-deacetylase inhibitor, has demonstrated achievement of a 27% ORR with 4% of CR in a heavily pretreated population of patients with relapsed/refractory cHL that had already failed a median number of four lines of therapy [35]. The time to response was 2.3 months, and the median duration of response was 6.9 months. Median PFS was 6.1 months, and the estimated 1-year OS was 78%. Panobinostat was, in general, a very well-tolerated drug, thrombocytopenia being the most frequent side-effect in 9% of the patients. Panobinostat has been combined with the ICE regimen in a recent Phase II study in which the combination was compared with ICE alone in this subgroup of patients [36]. The combination showed significantly higher CR in the experimental arm, but hematological toxicity was also higher.

### 3.3. JAK2 Inhibitors

Amplification of 9p24.1 is frequently seen in HL; this leads to the upregulation of JAK2, whose constitutive activation has been shown to be critical in HRS cell proliferation and survival [37,38]. A potent inhibitor of JAK1/2, ruxolitinib has been evaluated in a Phase II trial of patients with relapsed/refractory cHL [39]. In spite of the fact that ruxolitinib showed some anti-tumor activity with a 9.4% ORR, the response was not durable—the median duration of response was of 7.7 months, and there was a median PFS of 3.5 months. The drug was fairly well-tolerated. Ruxolitinib does not seem to be, at least as a single agent, an attractive strategy for the treatment of relapsed/refractory cHL; however, it might have a role in combination with other agents.

### 3.4. CART Cells

CART cell strategies have also been tested in relapsed/refractory cHL patients. Ten patients with refractory disease, including seven patients that had previously been exposed to BV, were treated with CART cells that expressed the antigen-binding domain of CD30; lymphodepletion was performed with the combination of fludarabine and cyclophosphamide [40]. Six of the nine evaluable patients (67%) achieved a CR, and four patients experienced grade 1 cytokine release syndrome (CRS). In a larger Phase I/II trial that included 24 heavily treated patients (22 HL patients), 10 of 19 (53%) patients achieved a CR at 6 weeks after the CAR T cell infusion; lymphodepletion was performed with bendamustine/fludarabine [41]. Median PFS was 164 days with a median follow-up of 180 days, but was 389 days for those 14 patients that received the higher dose of 2 × 10^8^ CAR-Ts/m^2^. CRS developed in four patients, three patients were grade 1, and one patient was grade 2, who responded to tocilizumab. Seventeen patients developed skin rashes that resolved spontaneously in all. Despite promising results, the small number of patients included and the short follow-up limited the strength of the conclusions that can be drawn from the results.

### 3.5. Allogeneic Stem Cell Transplantation

Allogeneic stem cell transplantation (allo-HCT) remains the only curative strategy for patients with HL that relapse after auto-HCT, thanks to the use of a donor immune system to prevent relapse. Nevertheless, allo-HCT is still associated with significant non-relapse mortality (NRM) that is basically derived from the development of both acute and chronic graft versus host disease (GVHD), with severe opportunistic infections usually in the context of GVHD and potential impairment in quality of life usually being associated to the presence of chronic GVHD. Nevertheless, the advent of new agents has made the timing and role of allo-HCT less clear in the last few years. Disease status at transplantation plays the most important role in terms of relapse rates and survival functions. The HDR-Allo trial [6] is the only prospective Phase II clinical trial that evaluates reduced-intensity allo-HCT. Ninety-two patients were enrolled; NRM was 15% at one year, and although 4-year PFS and OS rates were of 18% and 41%, respectively in the global population of patients, they went up to 40% and 60% in those patients that were allografted with chemosensitive disease. Haploidentical-HCT with post-transplant cyclophosphamide is now eventually being regarded as a standard procedure in HL patients that do not have an HLA identical sibling donor or an eventually matched, unrelated donor; it reduces GVHD incidence, is able to prevent rejection of the graft, and does not ablate the graft-versus-lymphoma effect. In spite of all these potential advantages, haploidentical-SCT is still associated to significant side effects, and NRM, although eventually reduced, is not absent. Both BV and checkpoint inhibitors are nowadays widely used before allo-HCT with the objective of achieving deeper clinical responses before the procedure [7,8,9], thus allowing a larger proportion of patients to be transplanted in CR or, at least, with chemosensitive disease. In this sense, the long-term outcome of allo-HCT could eventually be improved by the prior use of new agents.

BV has been largely used as a bridge to allo-HCT in several studies. Chen et al. [42] demonstrated that the use of BV before allo-HCT was able to significantly reduce the relapse rate after transplantation (23.8% vs. 56.5%) when compared to a group of patients that were bridged to transplant with conventional chemotherapy strategies; this reduced relapse rate was translated into a better 2-year PFDS in the same group of patients (59.3% vs. 26.1%). To the question of whether patients have to be allografted after achieving a CR with 3–4 cycles of BV or whether they could eventually continue to receive up to 16 cycles of the drug, or when disease progression is documented, provided a salvage treatment when an anti-PD1 agent is available [8,9], there is not a clear answer.

Because of its effectiveness, anti-PD1 agents have eventually been considered as the best option for bridging patients into allo-HCT. Nevertheless, the correct timing for an allogeneic transplant in patients receiving anti-PD1 agents is still an unsolved question. Should allo-HCT be performed in those patients that achieve a CR after therapy or at least PR or stable disease or, on the contrary, should we delay transplant until patients’ progress under anti-PD1 agents have provided an alternative regimen or a clinical trial is available? What we already know is that efficacy and safety of allo-HCT can eventually be modified by the prior use of CPIs, as a consequence of their immunomodulatory effect and their prolonged clinical activity. Residual PD1 inhibition may increase the allogeneic T-cell response, which in turn might be translated into both an augmented graft-versus-lymphoma effect and an enhanced incidence of GVHD, along with other immune-mediated complications [43]. Consensus recommendations for the treatment with CPIs in the allo-HCT setting based on the available data and experience have recently been published [44].

### 3.6. Radiotherapy

Radiotherapy (RT) has been widely demonstrated to be an efficient agent in the local control of relapsed/refractory cHL [45]. On the other hand, those patients that relapse with dissemination are unlikely to benefit from adjuvant RT. In this sense, the use of RT for relapsed/refractory disease after auto-HCT can be considered in different potential scenarios.

Palliative RT is an important approach for patients with relapsed/refractory HL that have no other systemic treatments left, who are not fit enough for more aggressive approaches, or who need time to recover from previous toxicities before they can receive additional systemic treatment [46].

Irradiation of localized persisting lesions after auto-HCT can be considered a valid approach for those patients with lesions in metabolic PR, in order to achieve minimal residual disease [46,47,48].

Nowadays, with the use of novel agents (target therapies and immunotherapies) in the relapse setting, RT is restricted for those patients with radiation-naive and localized disease, to try to avoid the exposition to systemic treatments and its toxicities. The combination of novel agents with RT is still under investigation. RT has a known effect in modulating immune responses, and its combination with checkpoint inhibitors is currently under investigation (“abscopal effect”) [49]. There are a number of currently ongoing prospective clinical trials in HL patients that will answer the question of its benefits, optimal schedule and dose of radiation, as well as some combinations that could potentially be fatal, such as the combination of anti-PD1 antibodies and mediastinal RT for the lung [50].

## 4. Conclusions

Although the outcome of patients with cHL that relapse or progress after auto-HCT has significantly improved after the introduction of both BV and checkpoint inhibitors, the curative capacity of these two strategies is almost non-existent. Nowadays, we have quite an impressive arsenal of new agents with different mechanisms of action; the intelligent combination of these different drugs will eventually improve not only the mCR rate, but also PFS, which at the end of the day, would eventually translate into the long-term curability of these patients (Appendix A). Hopefully, the introduction of new drugs in the first-line setting and the improvement of first-line salvaging strategies will significantly improve the long-term results of auto-HCT, and we will assist in the significant decrease of the percentage of patients that will need salvaging strategies because of autologous transplant failure (Appendix A).

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
