# Peer review of "Treatment of Hodgkin Lymphoma Relapsed after Autologous Stem Cell Transplantation"

_jcm, 2020, doi:10.3390/jcm9051384_

Round 1

Reviewer 1 Report

Suggestions for Authors:

Page 1, line 28: delete “advanced stage”.

Page 1 , line 33: write “for the majority of relapsed cHL patients...” instead of “for relapsed cHL patients….”

Page 2, line 66: write “durable CR and long term disease control without aditional consolidation treatment” instead of “durable CR that could spare additional consolidation treatment for long-term disease control”.

Page 3, line 97: Write “DCR” (disease control rate) instead of “ORR” (overall response rate).

Page 3, line 128: delete “the”.

Page 3, line 132: delete “relapsed”.

Page 4, line 173: write “the combination of bendamustine…..” instead of “the chemo-free approach...”. Bendamustine is chemotherapy.

Page 4: regarding the role of bendamustine monotherapy, data from the first phase II trial should be added (Moskowitz AJ, et al. J Clin Oncol 2013; 31: 456-60). In this trial, bendamustine was active (ORR 53%, CR 33%) but the median duration of response was only 5 months and less than 20% of patients remained without progression one year after treatment, despite the fact that 5 patients (20% of those eligible) proceeded to alloSCT after treatment with bendamustine.

Page 5 and 6 : Allogenic stem cell transplantation.

After the first sentence regarding allogenic stem cell transplantation (lines 211-213) “Allogeneic stem cell transplantation (allo-HCT) remains the therapeutic modality with the highest chance of cure for patients relapsing after auto-HCT with the use of a donor immune system to prevent relapse” the results of the only trial with intent to treat results should be remember in this article. The HDR-Allo study was a prospective trial of allogeneic stem cell transplantation with reduced intensity conditioning in patients with relapsed and refractory HL (Sureda A, et al. Haematologica 2012; 97: 310-317).  Ninety-two patients with a median age of 28 years were enrolled, and 78 with at least stable disease to salvage chemotherapy underwent an allograft. The nonrelapse mortality rate was 15% at 1 year. For the 92 included patients, the estimated 4-year PFS and OS rates were 18% and 41%, respectively”.

The authors should made some mention regarding the mortality and morbidity of haploidentical transplant.

Page 6: Conclusions

Line 247: write “for a selected subgroup of patients…..” instead of “for this subgroup of patients...”

Author Response

Dear reviewer,

Thank you very much for your valuable review.

  • Page 1, line 28:  “advanced stage” has been deleted.
  • Page 1 , line 33: we have chand “for the majority of relapsed cHL patients...” instead of “for relapsed cHL patients….”
  • Page 2, line 66: we have changed to “durable CR and long term disease control without aditional consolidation treatment” instead of “durable CR that could spare additional consolidation treatment for long-term disease control”.
  • Page 3, line 97: We have changed “DCR” (disease control rate) instead of “ORR” (overall response rate).
  • Page 3, line 128: we have deleted “the”.
  • Page 3, line 132: we have deleted “relapsed”.
  • Page 4, line 173: we have changed “the combination of bendamustine…..” instead of “the chemo-free approach...”. Bendamustine is chemotherapy.
  • Page 4: we have included the information of the manuscript of Moskowitz AJ, et al. J Clin Oncol 2013; 31: 456-60.
  • Page 5 and 6 : we have re-writen the Allo-HCT paragraph, including the manuscript of "Sureda A, et al. Haematologica 2012; 97: 310-317". We have done mention regarding the mortality and morbidity of haploidentical transplant.
  • Page 6: Conclusions: Line 247, we have changed “for a selected subgroup of patients…..” instead of “for this subgroup of patients...”

Reviewer 2 Report

I read with interest this review manuscript by Drs Domingo-Domenech and Sureda, focused on the treatment options of Hodgkin lymphoma relapsed/refractory after autologous stem cell transplantation. The authors provided a comprehensive review of the systemic agents available to date.

I have only a major comment on a significant limit of this manuscript: the authors did not address the role of radiation in this setting despite the numerous possibilities to integrate radiation in relapsed/refractory patients. Most relapsing patients, in fact, did not receive RT at any time of their previous treatments (in the first line but also in the “peri-transplant” period) for several reasons and, therefore, may benefit from radiation in the salvage setting after ASCT.  An evidence on the effectiveness of PD-1 inhibitors/RT combination is already available and many patients undergoing allo-HCT receive low dose total body irradiation as part of their conditioning regimen. Moreover, there is a huge debate on the possible “abscopal” effect of RT (even at low doses), particularly when integrated to CART cells. Lastly, RT may be offered as single agent in the palliative setting of patients unsuitable to more intensive treatments.

My suggestion is to dedicate a section to Radiotherapy in the manuscript in order to discuss all potential opportunities for the integration of local therapy in this challenging setting.

Author Response

Dear reviewer,

Thank you very much for your very important suggestion. We have included a section about the paper of radiotherapy in the relapse/refractory setting.

Reviewer 3 Report

The authors have presented a nice summary of the available therapeutic options for patients with cHL following ASCT.  I feel the manuscript would benefit from the following:

  1. A table of the agents, mechanisms of action, and outcomes
  2. 2. More importantly, an algorithm. What should the priority agent or regimen (might mention the BV-nivo combination of Herrera et al) in which setting.  When would you use BV vs a CPI?  For example, after the Aethera regimen, what would you use after a late relapse?  Such an approach would provide the reader some guidance rather than a list of avaialble drugs, some good, some not so.

Author Response

Dear reviewer,

Thank you very much for your review.

1.- We have added a table with the new agents and its combinations.

2.- We have also included our suggested algorithm for patients that relapse/refractory to Auto-HCT. The available drugs with its response rates have been included in the Table.

Reviewer 4 Report

Dear All 

This is an well written review article by Eva Domingo-Domènech  and Anna Sureda on treatment of cHL relapsed after autologous transplant.  This is an interesting topic with emerging new data that are presented in this review

Major comments:

  1. In part 3 a paragraph referred to prediction markers of response to check point inhibitors must be added. Some data exists about prediction of response in Nivo + BV combination ( CD30 levels MDR etc) I feel that another small paragraph referred to combinations of nivo ( nivo-ICE, Nivo-BV, PEM-ICE etc) as first salvage prior to auto must added too.

  1. I recommend a table with most important ongoing clinical trials to be included ( combinations or new agents ex sintilimab)

  1. Conclusions must be more Future perspectives instead of a summary of today’s evidences

Minor comments:

Some recommendations

Line 11 auto-HCT instead of  it

Line 16 allo-HCT instead of it

Line 36 change little informations

Line 37 auto-HCT (the same as before)

Line 46 after instead of with

Line 51 maybe who instead of that

Line 64 add Despite efficacy with BV monotherapy most patients required additional therapy within one year.

Line 72: add 12 patients had adverse events that lead to treatment discontinuation the most common being sensory neuropathy

Line 118 the case of Guillain – Barre syndrome should be mentioned

Line 121 change primary malignant cell…. To something  like ‘CHL the lymph node is composed of rare malignant cells (HRS) surrounded by inflammatory and immune cells infiltrate’

Line 124 change maintain the

Line 127 trimmed down ideal

Line 129 change frequently to universally

Line 135 add Notably even patients with SD had median PFS of 11 months and 1 year OS 96% similar to patients achieved PR or CR.  

Line 146 add of 151 responders 42,5% had response more than 24 months and 24.5% have ongoing responses.

Line 150 add AEs leads to treatment discontinuation in 6,7% of patients.

Line 160 a more extended reference to adverse events must added according to arms

Line 203 a line about maculopapular rash should be added

Line 235 caveats about toxicity GVHD, VOD etc I think should be mentioned

Author Response

Major comments:

  1. We have include data about the combination of nivolumab + brentuximab in second line, although the title of the review is treatments for relapse after auto-HCT, as well as a mention to other combination of checkpoint inhibidors as first rescue strategy.
  2. A table with new agents and its combinations has been included.
  3. Conclusions have been re-writen in order to explore more future perspectives.

Minor comments:

  • Line 11 we have changed to auto-HCT 
  • Line 16 we have changed to allo-HCT 
  • Line 36 we have changed to little informations
  • Line 37 we have changed to auto-HCT 
  • Line 46 we have changed to after
  • Line 51 we have changed to maybe who
  • Line 64 we have added Despite efficacy with BV monotherapy most patients required additional therapy within one year.
  • Line 72 we have added 12 patients had adverse events that lead to treatment discontinuation the most common being sensory neuropathy
  • Line 118 we have mentioned  the cases of case of Guillain – Barre syndrome
  • Line 121 we have changed the description of the HL lymph node
  • Line 124 we have re-writen this line
  • Line 127 have re-writen this line
  • Line 129 have re-writen this line
  • Line 135 we have added "Notably even patients with SD had median PFS of 11 months and 1 year OS 96% similar to patients achieved PR or CR"
  •   Line 146 we have added "of 151 responders 42,5% had response more than 24 months and 24.5% have ongoing responses."
  • Line 150 we have added "AEs leads to treatment discontinuation in 6,7% of patients".
  • Line 160 we have writen a more extended reference of AEs, only on the triple arm were this were really relevant
  • Line 203 we have added information about maculopapular rash
  • Line 235 we have included the caveats about toxicity GVHDand VOD

Round 2

Reviewer 2 Report

The authors addressed the role of RT in HL relapsed/refractory after ASCT.

Nothing more to add from my side.

Reviewer 3 Report

None